# Adhesive Performance of Acrylic Pressure-Sensitive Adhesives from Different Preparation Processes

**DOI:** 10.3390/polym13162627

**Published:** 2021-08-07

**Authors:** Irene Márquez, Núria Paredes, Felipe Alarcia, José Ignacio Velasco

**Affiliations:** 1Applications Department, Lubrizol Advanced Materials, Camí de Can Calders, 13, 08173 Sant Cugat del Vallès, Spain; irene.marquez@lubrizol.com (I.M.); nuria.paredes@lubrizol.com (N.P.); 2Poly2 Group, Department of Materials Science and Engineering, Universitat Politècnica de Catalunya (UPC BarcelonaTech), ESEIAAT, Carrer de Colom, 11, 08222 Terrassa, Spain; felipe.alarcia@hotmail.com

**Keywords:** acrylic pressure-sensitive adhesives, emulsion polymerization, glass bottle labels, post-polymerization blending

## Abstract

A series of pressure-sensitive adhesives (PSAs) was prepared using a constant monomeric composition and different preparation processes to investigate the best combination to obtain the best balance between peel resistance, tack, and shear resistance. The monomeric composition was a 1:1 combination of two different water-based acrylic polymers—one with a high shear resistance (A) and the other with a high peel resistance and tack (B). Two different strategies were applied to prepare the adhesives: physical blending of polymers A and B and in situ emulsion polymerization of A + B, either in one or two steps; in this last case, by polymerizing A or B first. To characterize the polymer, the average particle size and viscosity were analyzed. The glass transition temperature (Tg) was determined by differential scanning calorimetry (DSC). The tetrahydrofuran (THF) insoluble polymer fraction was used to calculate the gel content, and the soluble part was used to determine the average sol molecular weight by means of gel permeation chromatography (GPC). The adhesive performance was assessed by measuring tack as well as peel and shear resistance. The mechanical properties were obtained by calculating the shear modulus and determination of maximum stress and the deformation energy. Moreover, an adhesive performance index (API) was designed to determine which samples are closest to the requirements demanded by the self-adhesive label market.

## 1. Introduction

Pressure-sensitive adhesives (PSAs) have been in wide use since the late 19th century. They have achieved their popularity because they can adhere strongly to a wide variety of substrates at room temperature with the application of slight pressure for short time. Normally, the adhesives form the adhesion bond in liquid state and, by chemical reaction, a change in the temperature, UV irradiation, or another change, they pass to solid state, which is when they are tested. PSAs differ from others because they stick to a great variety of surfaces without any chemical reaction, only with light pressure [1]. They are widely used as, among others, surface protection films, a component of pressure-sensitive tapes, labels, notepads, skin-contact adhesive platform in medical plasters, and so on [2].

All commercial PSAs are based on polymers, mainly stemming from six families: rubbers (synthetics or naturals), polyesters, polyethers, silicones, polyurethanes, and acrylics. The remaining ones are based on esters of acrylic and substituted acrylic acids and present a great performance-cost and great relation qualities [3]. They are characterized by having high transparency; no color; low toxicity; high resistance to oxidation, UV, organic solvents, temperature, and plasticizers; and very good peel adhesion on polar surfaces.

Although this type of PSA can be obtained by different polymerization processes, emulsion polymerization shows several advantages compared with bulk and solvent polymerization. In addition to offering environmental safety owing to the use of water instead of organic solvents, the polymerization process takes places inside of the polymer particles, allowing producing polymers with a high solid content, high polymerization rates, and low viscosity [4].

In addition to tack, which represents the ability of the PSA to form bonds with a substrate with a brief contact under slight pressure, they must have a certain peel and shear resistance. Peel resistance is defined as the force required to remove a tape from a test panel and shear resistance is defined as the capacity of the PSA tape to remain adhered under constant load. Tack and peel resistance are associated with adhesion forces and shear resistance is associated with cohesion forces. Adhesion and cohesion forces are the forces that intervene in the bond. Adhesion is the attraction of two different substances resulting from intermolecular forces between the materials. However, cohesion involves only the intermolecular attractive forces within the adhesive.

The intermolecular forces acting in both adhesion and cohesion are primarily van der Waals forces. The balance between these ones determines the properties and quality of the adhesive bond [5,6]. Normally, increasing the properties’ associated with adhesion tends to decrease cohesion as they are opposing forces. For this reason, it is not possible to obtain the maximum value of both at the same time, but it is possible to reach a balance [7].

This balance between these properties is greatly affected by the process employed to obtain the polymer, regardless of whether it is the same formulation. Normally, this goal can be achieved by modification of the chain transfer agent concentrations [8], modification of reactor conditions [9], or polymerization using different catalysts [10]. An alternative option would be to mix two resins of different adhesive properties. Fonseca et al. [11] compared the final adhesive properties of in situ polymerized samples with those of prepared by physical blend after their polymerization. The results showed that gel content of in situ polymers was higher than that obtained by the blend of polymers and, in general, the adhesive properties were higher in the first case. Tobing et al. [12] studied the adhesive performance of high-gel containing latices and gel-free latex blended at various weight ratios. The results showed that the shear resistance was higher in the emulsion blend than in the individual latexes. Jovanovic et al. [13] compared the results between the conventional emulsion and the miniemulsion polymerization, where the last one showed a higher control of the molecular weight distribution. In this case, a high tack, moderate peel resistance, and low shear adhesion were obtained. They suppose that the presence of long chains alone was not enough to improve the shear.

In the field of self-adhesive labels, PSAs have acquired great importance. Their features allow labels to be built and stored in reels, improving their labelling processes [1]. As the years go by, label manufacturers claim better properties in the adhesives supplied to them owing to the great competition in the market. According to the market, currently, PSAs designed for this type of application should present peel, tack, and shear values of 10 N/25 mm, 10 N, and 24 h, respectively. However, it is very difficult to meet all three requirements at the same time because, normally, when the adhesive properties (peel and tack) increase, the cohesive properties (shear) decrease, and vice versa.

The general aim of this work is to develop PSAs for glass bottle labels that meet the requirements currently demanded by the market. As mentioned, this is very complicated. In previous articles [14,15], an attempt has been made to arrive at this balance by modifying the composition of the polymer. However, the desired values have not been reached. For this reason, this study has attempted to reach this objective by modifying only the process for obtaining the polymer, keeping the monomeric composition constant. To find this optimal balance in the adhesive properties, between tack, peel resistance, and shear resistance, the monomeric composition was a 1:1 combination of two different water-based acrylic polymers, A and B. Polymer A was designed with a high content of acrylic acid (AA) in order to obtain high molecular weights that would favor the attainment of high values of shear resistance. Instead, polymer B was designed with a low AA content and a chain transfer agent (tert-dodecyl mercaptan) was added to the formulation in order to obtain low molecular weights that gave good peel resistance and tack results. Both A and B were copolymerized with butyl acrylate and acrylonitrile, as shown in Figure 1. This type of polymerization takes place in a random way [15].

Four methods were carried out to obtain the polymers: by physical blend of polymers after their polymerization; by polymerization in situ in one; and by polymerization in situ in two steps, where in each step, one of these formulations (A or B) was synthetized and two different polymerization orders were tested, as shown in Figure 2.

To determine the differences between them, the typical physico-chemical and adhesive and mechanical properties were investigated. Moreover, an adhesive performance index (API) was designed to determine which samples are closest to the requirements demanded by the self-adhesive label market.

## 2. Materials and Methods

Acrylic acid (AA) and n-butyl acrylate (n-BA) provided by BASF (Ludwigshafen, Germany), as well as acrylonitrile (ACN) provided by IMCD Benelux B.V. (Amsterdam, The Netherlands), were used as comonomers in the polymerization. Tert-dodecyl-mercaptan (TDM) provided by Chevron Phillips Company LP (Tessenderlo, Belgium) was used as a chain transfer agent. The anionic polymerizable surfactant (S), Maxemul^TM^ 6112, based in a modified alcohol ether phosphate, provided by Croda (Mill Hall, PA USA), was also used in the polymerization. Ammonium carbonate ((NH_4_)_2_CO_3_) provided by BASF (Ludwigshafen, Germany) was used as a buffer and ammonium peroxide sulphate (I) supplied by United Initiators (Pullach, Germany) was used as a thermal initiator. 

A combination of tert-butyl hydroperoxide (TBHP), provided by Pergan (Bocholt, Germany), and sodium formaldehyde sulfoxylate (Bruggolite^®^ E01), from Brüggemann KG (Heilbronn, Germany), were used as a redox system to reduce free monomer at the end of the polymerization. A 25% ammonia solution, provided by Barcelonesa drugs and chemicals (Cornellà del Llobregat, Spain), was used to neutralize the adhesives. Tetrahydrofuran (THF) at 99%, provided by Merck (Hohenbrunn, Germany), was used as a solvent to determine the gel content as it is the most suitable solvent to dissolve the acrylic polymers studied here. Accordingly, it was also used as the mobile phase in the gel permeation chromatography (GPC) measurements. 

For adhesion tests, a polyethylene terephthalate (PET) of 12 µm provided by Polinas (Manisa, Turkey) with corona treatment as a surface activation treatment [16] and Tintoretto qesso ultraWS^TM^ paper provided by Arconvert (Sant Gregori, Spain) were used as tapes to perform the tests.

### 2.1. Sample Preparation

Samples were prepared by two different methods: by emulsion polymerization and by physical blend of polymers after their polymerization.

#### 2.1.1. Emulsion Polymerization

Samples A, B, 1sp-AB, 2sp-AB, and 2sp-BA were prepared by emulsion polymerization. Here, 2 kg of each polymer was prepared at 55% of solid content, adjusting the quantity of water to keep this rate constant. The polymerizations were carried out by a semi-batch emulsion process in a 2.5 L glass reactor at 82 °C with mechanical stirring at 100 rpm. The initial charge in the reactor consisted of 0.3 parts of (NH_4_)_2_CO_3_ per 100 parts by weight of monomer (i.e., 0.3 phm), 0.1 phm of emulsifier, and half of the total water. 

After heating and purging the reactor with N_2_, the thermal initiator was added followed by the pre-emulsions composed by the emulsifier, monomeric system (Table 1), and the remaining water. 

In the case of the polymerization in situ in one step (samples A, B, and 1sp-AB), after adding the thermal initiator, the pre-emulsion was added at a constant rate over 3 h.

However, in the case of the polymerization in situ in two steps (samples 2sp-AB and 2sp-BA), the formulation for each step was designed at 27% of solid content to finally obtain a polymer at 55% solids. After adding the thermal initiator, the first pre-emulsion (Step 1) was added at a constant rate for 1.5 h. Then, the other thermal initiator was added followed by the second pre-emulsion (Step 2).

Once the pre-emulsions feed was completed, to consume the residual monomer, two shots of 1.0 phm of thermal initiator were added (each one was allowed to react for 1 h). Later, the reactor was cooled down to 57 °C and a redox couple of TBHP/Bruggolite^®^ E01 (0.2 phm/0.3 phm) was added to consume the residual monomer. Post-polymerization was allowed to take place over 4 h. Gas chromatography analysis indicated that the free monomer concentration was lower than 750 ppm.

#### 2.1.2. Physical Blend of Polymers

Here, 2 kg of sample b-AB was prepared by mechanical mixing of A and B (1:1) at 55% of solid content in water. It was carried out in a 2.5 L glass reactor at 25 °C with stirring at 100 rpm during 30 min.

### 2.2. Latex Characterization

After the emulsion polymerization, the latex obtained was cooled at room temperature and then filtered through a 150 µm filter, and then analyzed to determine their physical-chemical characteristics. Prior to the characterization, all the synthetized adhesives were normalized by adjusting the pH to 7.5 and the solid content to 50 wt. % by neutralizing with ammonia solution (12.5%) and adding deionized water, as necessary.

The average particle size was measured by dynamic light scattering (DLS) with a Zetasizer Nano Series instrument. Samples were prepared by diluting the polymer in deionized water and analyzed at 25 °C, using a detector with a 90° angle. The viscosity was determined at 25 °C using a programmable Brookfield DV-II+ Pro rotational viscometer for low viscosities.

The Tg was determined experimentally by differential scanning calorimetry (DSC) using the equipment DSC 1, STARe, calibrated with an indium and zinc standard. Samples of about 20 mg were initially placed in crucibles and dried in an oven at 80 °C for 5 h to obtain dry test samples of about 10 mg. These samples were firstly heated at a rate of 20 °C/min from 25 °C to 200 °C and cooled to −65 °C at 20 °C/min. After stabilization for 15 min at −65 °C, the second heating was carried out at 20 °C/min up to 200 °C. The Tg value was obtained from the second heating curve.

The gel content was defined as the polymer fraction insoluble in THF at 70 °C. To obtain this polymer fraction, it is necessary to form macromolecules with molecular weight higher than 7 × 10^6^ g/mol, according to the literature [17]. It was determined after Soxhlet extraction for 24 h. The studied samples are thermoplastic polymeric materials with presence a gel fraction distributed in the form of small gel domains in the bulk. According to the literature, these gel domains consist of cross-linked polymer and entangled high molecular weight polymer chains [18].

This fraction was dried in an oven at 60 °C for 24 h to determine the gel content using Equation (1), where W_1_ represents the initial weight of the filter, W_2_ is the weight of the filter with the dry polymer, and W_3_ is the final dry weight of the filter after extraction [19].
(1)$$\mathrm{Gel}\;\mathrm{content}\;(\%)=\frac{W_{3}-W_{1}}{W_{2}-W_{1}}\;100$$

The average molecular weight of the soluble polymer fraction (Mw) obtained was determined by GPC. The samples taken out from the Soxhlet were first dried and resolved in THF to achieve a concentration of about 0.01 g/mL, and 10 μL was injected into the GPC instrument. The device consisted of a Waters 2414 refractive index detector and a Waters e2695 Separation Module equipped with one lineal column of Agilent PL Gel Mixed-C of 5 µm of 7.8 × 300 m, followed by a lineal column of Styragel HR5E of 5 µm of 7.8 × 300 m and one monopore column of Styragel HR 4 7.8 × 300 mm. Chromatograms were obtained at 40 °C using a THF flow rate of 1 mL/min as the continuous step. The equipment was calibrated using polystyrene standards (PSs) and, therefore, the average molecular weight referred to PS.

### 2.3. Adhesion Tests

The adhesive properties were determined through peel as well as tack and shear resistance test. Using a motorized laboratory coater, RK K Control Coater provided by Lumaquin S.A. (Montornès del Vallès, Spain), equipped with a bar of 50 µm, 50 g/m^2^ of polymer was applied onto the tapes, which were subsequently dried in the oven for 1 min at 100 °C, leaving a layer of polymer of approximately 25 g/m^2^. Standard sized tapes were cut for each type of test.

The peel resistance, defined as the force required to remove a tape from a test panel, was evaluated by means of the 180° peel test after 24 h from the tape application. Paper and PET tapes of 275 × 25 mm^2^ were applied onto glass panels. A Zwick/Roell Z 2.5 tensioner was used at a constant speed of 300 mm/min. The average force to remove the tape and the failure mode were recorded [20].

Tack is the capacity of the adhesive to form bonds with a substrate with a brief contact under slight pressure. Tack was determined by the loop tack test with an AT1000 tensile tester equipment. A loop was formed with a paper/PET tape of 175 × 25 mm^2^ and held with the upper clamp. A controlled contact was made at a constant speed of 300 mm/min onto glass panels. The maximum force required to peel off the tape from the panel and the failure mode were recorded [21,22].

The shear resistance is defined as the capacity of the PSA tape to remain adhered under constant load applied parallel to the surface of the tape and substrate. This test involves applying a standard area of paper/PET tape of 25 × 25 mm^2^ on a panel of stainless steel to 2° from the vertical and holding 1 kg until failure. The average time the tapes take to shear from the test panel was recorded [23].

Dynamic shear tests were performed at 5 mm/min with a Zwick/Roell Z 2.5 machine on PET tapes adhered on untreated steel panels at 25 °C with a contact area of 2.5 × 2.5 mm^2^. Twenty minutes before the test, the test sample was pressed by means of a rubber roller with a mass of 2 kg four times, according the procedure [24]. The shear stress versus strain curves were recorded and the elastic modulus (G), the maximum stress (τ_m_) values, and the deformation energy up to failure (U) were determined.

The shear modulus was determined as the initial slope of the curve with a linear correlation coefficient (r^2^) being higher than 0.999 in all cases. The shear strength was determined as the maximum stress value in the test, and the deformation energy was calculated as the area under the curve up to the maximum stress value.

## 3. Results and Discussion

The average particle size and viscosity of the aqueous polymer solution (50% solids) as well as the Tg, gel content, and average sol molecular weight (Mw) of the polymer are summarized in Table 2.

The average particle size of samples b-AB and 1sp-AB, physical blend of polymers, and polymerization in situ in one step, respectively, were very similar, an intermediate value between the mean particle size of samples A and B. The viscosity value for sample b-AB was the mean value between A and B. Instead, sample F showed a viscosity similar to sample B. However, samples 2sp-AB and 2sp-BA obtained by polymerization in two steps showed lower average particle size and viscosity than the other samples. This was because new particles were formed in each step. The Tg results provided by DSC shown in Table 2 demonstrate that the differences found could not be considered significant because the composition was the same. However, if gel content versus Tg is represented (Figure 3), a trend can be seen.

Samples with a higher gel content displayed higher Tg values. This relationship is generally attributed to the effect of the polymer chain mobility. Those samples with more entanglements and crosslinked display less mobility, which was reflected in their Tg values [25,26].

The average molecular weight determined by GPC only corresponds to the fraction of the polymer solubilized in THF. Therefore, the values obtained should just be considered as the molecular weight of the polymer without the insoluble fraction in THF, that is, without the gel content, that cannot be analyzed by GPC. Samples A and 2sp-AB showed the highest gel content, 72% and 54%, respectively, and thus the highest molecular weight. In these cases, the values obtained by GPC of the Mw were not so representative, as the proportion of sample examined was very low.

The level of entanglement showed by the samples was due to the presence of carboxylic acid groups [27] and the reactions of backbiting and intermolecular chain transfer to polymer followed by termination by combination that take place during emulsion polymerization [28]. However, these effects were insignificant in sample B, owing to the presence of the chain transfer agent [29,30,31,32]. An amount of 0.2 phm of TDM was enough to avoid the gel content in this case. Samples 1sp-AB and 2sp-BA showed negligible values of gel content in comparation with samples A, b-AB, and 2sp-AB. Sample b-AB showed an intermediate value between samples A and B, as it was a mechanical mix between both. However, sample 2sp-AB showed a greater gel content than sample 2sp-BA. This could be because, in sample 2sp-BA, TDM was added in the first step. This meant that the TDM that did not react in the first step did so in the second, thus decreasing the gel formation in both steps. Instead, in sample 2sp-AB, it only intervened in the last step.

The adhesive properties (tack, peel resistance, and shear resistance) of polymers, tested on paper and PET tapes, were measured and summarized in Table 3.

The polymer obtaining process gave rise to samples with differing gel contents, and thus with different molecular weight distributions. These distinctions in molecular weight were responsible for the differences found in their adhesive properties. 

Both tack and peel resistance increase with increasing molecular weight until a maximum is achieved. Tack is influenced by the low-range molecular weight and peel resistance is known to be influenced by the middle-range molecular weight. The maximum is at a fairly low molecular weight and the transition of the mode of failure from cohesive to adhesive in peel test takes place in this region. Beyond the maximum of the tack and peel resistance, when an increase in the molecular weight causes a decrease of these properties, shear resistance increases. This rise with the growth of molecular weight up to the point where adhesion has decreased to such a low level that the adhesion failure takes place when a shear force is applied [7,32].

Three different types of failure were observed: cohesive, adhesive, and transfer failure. Cohesive failure took place when the adhesive remained on both sides of the joint during the peeling test. The adhesive samples made of mixtures of A + B polymers remained completely adhered with no residue on the support material, which is adhesive failure. On the contrary, when the opposite occurred, that is, the adhesive was completely transferred from the tape to the support, it was called failure by transfer. 

On paper tapes, all samples made from formulations A and B showed a rise in peel resistance values going from a transfer and cohesive failure, respectively, to an adhesive one. In this case, sample b-AB showed the highest value, followed by sample 1sp-AB. The reason for these increases found among these latexes could be that they present a greater amount of chains in these middle ranges of molecular weight. On the other hand, sample 2sp-AB showed tack values more similar to sample A, because the chain transfer agent only intervened in the last step. 

Otherwise, samples b-AB and 2sp-BA were more similar to sample B. In both cases, 0.1 phm of TDM was enough to obtain the same values of B, where 0.2 phm of TDM was used. However, sample 1sp-AB showed an increase of 70% with respect to sample B. Regarding the results obtained in the shear resistance test, in general, all samples showed very low values, except for sample A, owing its high AA content. However, sample 1sp-AB was the second sample with the highest value.

On PET tapes, all samples made from formulations A and B showed lower peel resistance values and showed adhesive failure. Sample B, which showed the highest peel resistance values, was the only one that showed a cohesive failure. The same effects were shown in the tack values. However, this time, the difference between B and samples b-AB and 1sp-AB was not that great. Regarding the results obtained in the shear resistance test, in general, all samples showed very low values, except for sample A. However, sample 2sp-AB was the second sample with the highest value, although it was 60% less than A.

The adhesive performance is closely related to the nature of the chosen tape. As paper is a porous substrate, the anchorage interface of adhesive–substrate is higher than the internal strength of the adhesive because a great proportion of the adhesive penetrates the matrix paper. However, this does not happen with PET substrate. For this reason, the peel and tack values were lower and shear resistance values were higher in PET.

Increasing the peel and tack values is a relatively easy process as blending two polymers will most likely increase the amount of low and medium molecular weight chains. However, because shear resistance is determined by the presence of long molecules, it should not be possible to increase this property by simple mechanical blending [13]. It would be necessary to add a crosslinking agent or remove chain transfer agents from the formula for typical polymer chain transfer reactions to occur. For this reason, the results show a significant increase in peel resistance and tack, but not in shear resistance. The same effects were reflected in the dynamic shear resistance results.

As shown in Figure 4, the shear elasticity modulus (G) showed small variations depending on the procurement strategy used. 

Samples B and 1sp-AB showed the lowest G owing to their low Mw. On the other hand, sample A and 2sp-AB showed the highest G values. This is because they were the samples with the highest gel content and, therefore, with the highest entanglement. This prevents the chains from having mobility, thus reducing the elasticity of the sample. In contrast, samples b-AB and 2sp-AB showed intermediate values of G.

The maximum shear stress recorded in the dynamic shear test (Figure 5) showed the same tendency as in G.

Samples A and 2sp-AB showed the highest values. In contrast, sample B showed the lowest values owing to its greater elasticity. Samples b-AB, 1sp-AB, and 2sp-BA showed values more similar to A than B. This was because all samples generated gel content, except B, thus the molecular weight of the latter was much lower, giving rise to a more flexible polymer than the rest.

Finally, the toughness of the adhesives was determined as the deformation energy up to the adhesive failure in the dynamic shear test and was calculated from the area under the force–displacement curve up to the maximum force value. As shown in Figure 6, sample A required more energy than the rest.

This sample was prepared with the highest AA content and the absence of TDM, which generated a high gel content in the final polymer. However, sample 2sp-AB showed a deformation energy three times lower than that in sample A. This was due to the effect of the chain transfer agent and the amount of acrylic acid, as previously commented. In contrast, samples B and 1sp-AB showed the lowest values.

To determine the samples that satisfy the currently requirement demanded by the market, an adhesive performance index (API) was defined, which takes into account the relative importance of the three adhesive properties in the performance of the adhesive. The polymer that shows a higher API value will be the one that is closest to the requirements. To develop the API, it was considered that the values required of peel resistance (PR_t_), tack (T_t_), and shear resistance (SR_t_) by the wine industries were 10 N/25 mm, 10 N, and 24 h, respectively. The API was obtained with the average value obtained of peel resistance (PR), tack (T), and shear resistance (SR) for each sample (s). As the differences in the shear resistance test were very large, the natural logarithm (ln) was applied to the values obtained. As a rule, the industry of labels generally gives much more importance to adhesive shear resistance than the other adhesive properties (i.e., tack and peel). In order to combine the relative importance of the three adhesive properties given by this industry to the PSA for labels, the API can be determined by means of the following Equation (2):
(2)$$\mathrm{API}=\mathrm{Cp}\;\cdot \frac{{\mathrm{PR}}_{s}-{\mathrm{PR}}_{t}}{{\mathrm{PR}}_{t}}+\mathrm{Ct}\;\cdot \;\frac{T_{s}-T_{t}\;}{T_{t}}+\mathrm{Cs}\;\cdot \;\frac{\ln({\mathrm{SR}}_{s})\;-\;\ln({\mathrm{SR}}_{t})}{{\ln(\mathrm{SR}}_{t})}$$
where Cp, Ct, and Cs are coefficients that quantify the relative importance of peel, tack, and shear resistance, respectively, in the adhesive performance. For application in labels for wine glass bottles, these coefficients are accepted to have values of 15%, 15%, and 70%, respectively. 

As shown in Figure 7, according to the values assigned to determine the API, the samples that most closely approximated the wine cellars’ requirement in the case of paper substrate were samples A and 1sp-AB.

In the case of sample A, the shear values increased almost 17 times, but its peel and tack values were less than 10 N/25 mm and 10 N, respectively. On the other hand, sample 1sp-AB showed the opposite case. It showed an increase of 100% in both peel and tack from what was expected, but the shear values were four times lower. In both cases, the API result is highly influenced by the shear values. 

On the other hand, in the case of PET substrate (Figure 8), the samples that most closely approximated the wine cellars’ requirement were samples A and 2sp-AB.

In both cases, the performance index result was highly influenced by the shear values, and their peel resistance and tack values were less than 10 N/25 mm and 10 N, respectively.

## 4. Conclusions

To obtain the ideal balance between the three adhesive properties most demanded in the field of PSA, different methods of obtaining the same formulation were tested. Two different polymers, one with a high shear resistance (A) and the other with a high peel resistance and tack (B), were used to design the base formulation. The designed formulation was a 1:1 combination of both polymers. The studied polymers were obtained by physical blend of polymers of both and by polymerization in situ in one and in two steps. Although it was the same formulation, great differences were found in terms of its final properties depending on the obtaining process. Regarding the physico-chemical properties, physical blend of polymers and one step polymerization showed similar results. In contrast, the two-step polymerization resulted in lower particle sizes and viscosities. Significant differences were found in the results of gel content and Mw, highly influenced by the presence of TDM and the amount of AA. The samples with a lower gel content displayed lower Tg values. In the case of paper substrate, all samples designed from A and B displayed increased peel values. Sample b-AB was the one with the highest value, followed by sample 1sp-AB, which also showed the highest tack values exceeding 70% of sample B. Regarding the shear values, none approached the values of sample A; however, sample 1sp-AB was the most effective of the designed samples. On the other hand, in the case of PET substrate, all samples designed from A and B showed peel and tack values lower than B and shear values lower than A. The dynamic shear test showed that the highest values of G, maximum shear stress, and deformation energy until failure were obtained with samples A and 2sp-AB owing to their high gel content.

From the results obtained, an API was carried out to evaluate if the samples gathered through the different processes were capable of meeting the wine cellar requirements. In paper substrate, samples A and 1sp-AB were the closest to the objective. However, for PET substrate, samples A and 2sp-AB were the closest. For both substrates, sample A was one of those selected owing to its high shear values, but showed slightly lower peel and tack values. In the case of the paper substrate, sample 1sp-AB exceeded more than 100% in terms of peel and tack values. However, its shear values were four times lower than required. On the other hand, in the case of the PET substrate, 2sp-AB showed slightly lower peel and tack values and double the required shear value.

## Figures and Tables

**Figure 1 polymers-13-02627-f001:**
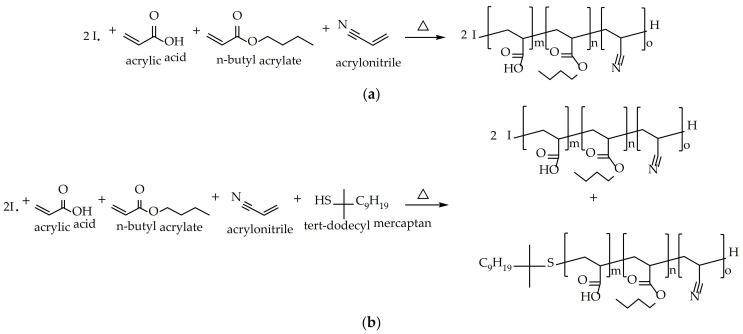
Reaction scheme of (**a**) sample A and (**b**) sample B.

**Figure 2 polymers-13-02627-f002:**
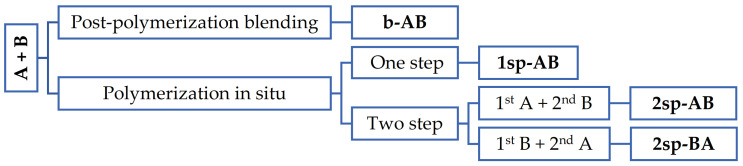
Scheme of the strategy used to obtain the samples. A and B are the base polymers designed to obtain high shear values and high peel and tack values, respectively. 1sp and 2sp are the number of steps, one and two, respectively.

**Figure 3 polymers-13-02627-f003:**
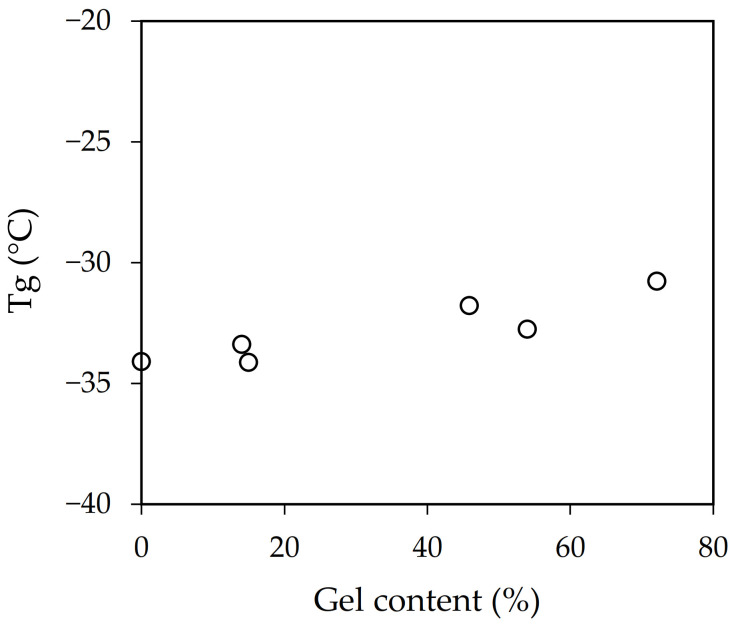
Influence of the gel content on the Tg values.

**Figure 4 polymers-13-02627-f004:**
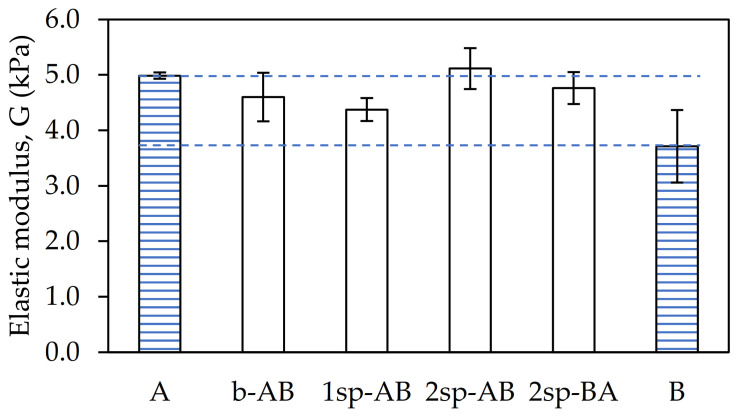
Shear modulus values for each adhesive sample.

**Figure 5 polymers-13-02627-f005:**
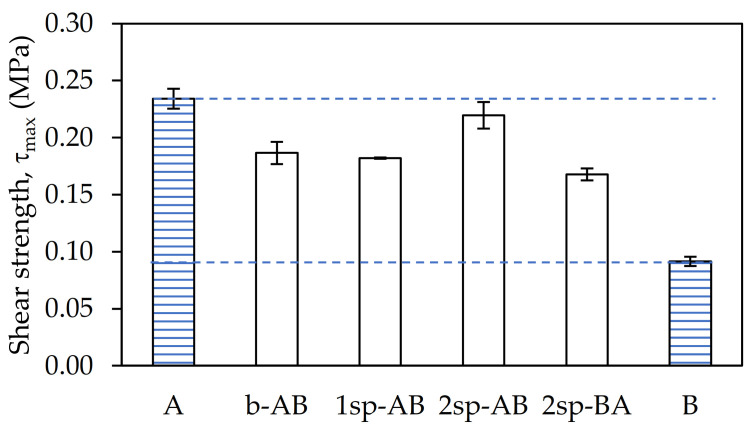
Maximum shear stress values for each adhesive sample.

**Figure 6 polymers-13-02627-f006:**
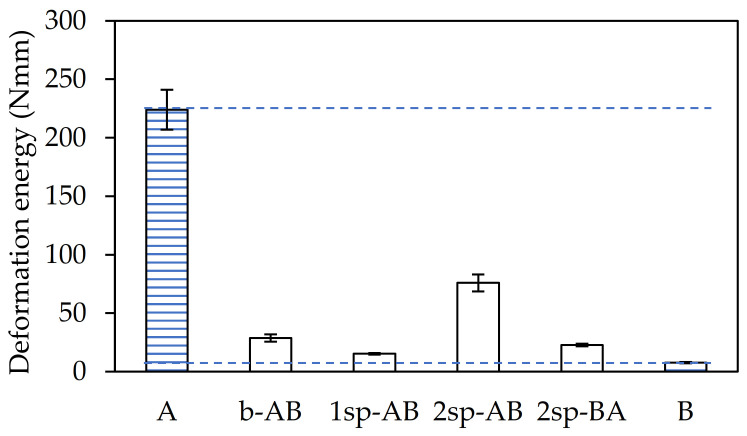
Deformation energy until failure for each adhesive sample.

**Figure 7 polymers-13-02627-f007:**
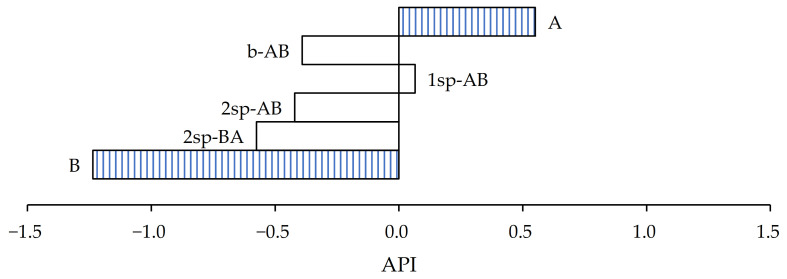
Adhesive performance index of samples for paper substrate.

**Figure 8 polymers-13-02627-f008:**
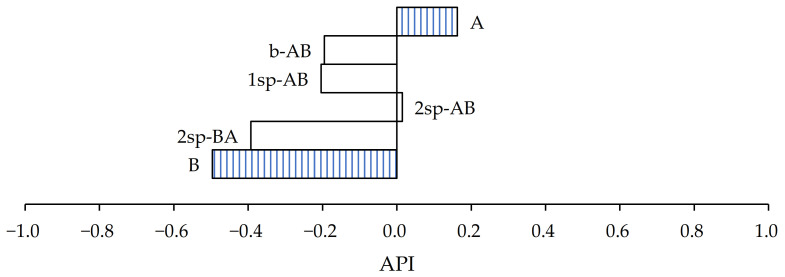
Performance index of samples for PET substrate.

**Table 1 polymers-13-02627-t001:** Initiator (I), surfactant (S), chain transfer agent (TDM), and monomer composition (phm): acrylic acid (AA), n-butyl acrylate (n-BA), and acrylonitrile (ACN).

Sample	Step 1						Step 2					
	AA	n-BA	ACN	TDM	S	I	AA	n-BA	ACN	TDM	S	I
A	3.00	91.0	6.0	0.0	1.2	0.5						
B	0.50	93.5	6.0	0.2	1.2	0.5						
b-AB	1.75	92.3	6.0	0.1	1.2	0.5						
1sp-AB	1.75	92.3	6.0	0.1	1.2	0.5						
2sp-AB	1.50	45.5	3.0	0.0	0.6	0.25	0.25	46.8	3.0	0.1	0.6	0.25
2sp-BA	0.25	46.8	3.0	0.1	0.6	0.25	1.50	45.5	3.0	0.0	0.6	0.25

**Table 2 polymers-13-02627-t002:** Physico-chemical properties.

Sample	Particle Size (nm)	Viscosity (cp)	Tg (°C)	Gel Content (%)	Mw (g/mol)
A	207	165	−30.8	72	476,516
B	270	81	−34.1	0	231,126
b-AB	236	124	−31.8	46	446,276
1sp-AB	234	75	−33.4	14	276,767
2sp-AB	164	23	−32.8	54	993,126
2sp-BA	172	49	−34.1	15	505,048

**Table 3 polymers-13-02627-t003:** Adhesive properties determined on both paper and PET tapes. The failure mode is indicated in parenthesis.

	Paper Tape			PET Tape		
	Peel Resistance (N/25 mm)	Tack (N)	Shear Resistance (h)	Peel Resistance (N/25 mm)	Tack (N)	Shear Resistance (h)
A	7.7 ± 0.2 (Transfer)	7.6 ± 0.2	402 ± 50	4.9 ± 0.3 (Adhesive)	3.4 ± 0.5	112 ± 5
B	19.6 ± 0.7 (Cohesive)	11.6 ± 1.3	0.1 ± 0.0	21.0 ± 0.8 (Cohesive)	10.0 ± 0.5	1.2 ± 0.2
b-AB	26.0 ± 0.8 (Adhesive)	13.0 ± 1.4	1.2 ± 0.2	5.2 ± 0.6 (Adhesive)	9.2 ± 0.7	14.4 ± 0.6
1sp-AB	24.6 ± 0.8 (Adhesive)	19.7 ± 0.3	6.2 ± 0.4	8.1 ± 0.4 (Adhesive)	8.7 ± 0.5	11.8 ± 0.7
2sp-AB	21.9 ± 0.6 (Adhesive)	8.0 ± 0.5	1.8 ± 0.9	4.9 ± 0.4 (Adhesive)	6.3 ± 0.6	47.0 ± 0.9
2sp-BA	22.8 ± 1.0 (Adhesive)	11.4 ± 0.2	0.8 ± 0.0	7.0 ± 0.5 (Adhesive)	6.5 ± 0.3	6.7 ± 1.0

## Data Availability

Not applicable.

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
