# Peer review of "Adhesive Performance of Acrylic Pressure-Sensitive Adhesives from Different Preparation Processes"

_polymers, 2021, doi:10.3390/polym13162627_

Round 1

Reviewer 1 Report

Review in the attachment 

Author Response

Dear Reviewer #1,

Thank you very much for your time and comments on our manuscript. We have addressed the suggestions given in your report:

English language and style has been checked.

The introduction, description of the experimental methods and results have been revised and improved according to the reviewers’ comments in the revised manuscript.

All the changes have bee market in the revised manuscript in red color.

Reviewer 2 Report

The authors present a readable paper regarding the preparation strategies of acrylic acid-based pressure sensitive adhesives. I found the manuscript fluent and the techniques well explained, the introduction is well done and the relative literature coherent. Conclusions and experimental part are consistent and without any major imprecisions. I have some minor comments:

-keywords: resolve the acronym PSA plus presenting only the word "PSA" and add another one as you have 5 slots more there are better it is for people seraching for your paper

-title : please resolve the acronym PSA, not all the community know at a glance about what it is referred

-line 16: please rephrase “…in two steps, where each step was synthetized one of these formulations.” It is too difficult “…in two steps, either by polymerizing A or B first.”

- line 41: a list has to be consistent, they are characterized by having colorless…; …by having no color;…

-line 43, 48: specify “solvents”; water is a solvent.

-caption of figure 1: the captions have to be self-standing, by reading alone you should get the meaning, please specify the what is sp, A, B, etc.

-line 127: specify and/or give references about the corona treatment

Line 178/185…: use the same symbol for degree Celsius º or ° (with or without the underscore).

Author Response

Dear Reviewer #2,

First, thank you very much for your comments. We have read them carefully and have tried to address all the recommendations proposed. Here is a brief summary of what has been changed according to your comments.

Comment 1:keywords: resolve the acronym PSA plus presenting only the word "PSA" and add another one as you have 5 slots more there are better it is for people searching for your paper.”

It has been changed and another keyword has been added: post-polymerization blending.

Comment 2:title: please resolve the acronym PSA, not all the community know at a glance about what it is referred.”

The title has been changed accordingly.

Comment 3:line 16: please rephrase “…in two steps, where each step was synthetized one of these formulations.” It is too difficult “…in two steps, either by polymerizing A or B first.”

It has been rephrased.

Comment 4:line 41: a list has to be consistent, they are characterized by having colorless…; …by having no color;…”

It has been changed accordingly.

Comment 5:line 43, 48: specify “solvents”; water is a solvent.”

It has been replaced by “organic solvents”.

Comment 6:caption of figure 1: the captions have to be self-standing, by reading alone you should get the meaning, please specify the what is sp, A, B, etc.”

The meaning of A, B, 1sp and 2sp has been given in the title of the figure.

Comment 7:line 127: specify and/or give references about the corona treatment.”

One additional reference has been added in that section for readers who need more information on the subject.

Comment 8:Line 178/185…: use the same symbol for degree Celsius º or ° (with or without the underscore).”

It has been changed.

Reviewer 3 Report

PSA (Pressure Sensitive Adhesive) is a type of very efficient and safe adhesive. This substance binds to almost any material with light pressure applied. PSA binders are very popular, due to these properties, as well as the absence of unpleasant odors and high resistance to extreme temperatures.

This type of adhesives form a bond when both substrates are pressed down together. As the name implies, the degree of connection is influenced by the applied force. Since the adhesive remains tacky, it has high bond strength but, on average, lower cohesive strength. Therefore, the adhesive is ideal for applications where it is necessary to re-connect or reposition parts.

The reviewed article is written correctly. It is an interesting approach to the  PSA adhesives topic.

A series of PSA adhesives were prepared using a constant monomer composition and various preparation methods, as part of the research work. Adhesives were prepared by physically mixing polymers and their polymerization and by in situ polymerization in one and two stages, where one of the preparations was synthesized at each stage. The monomer composition was a combination of two different polymers, one with high shear strength (A) and the other with high peel strength and adhesion (B).  Average particle size and viscosity were analyzed to characterize the polymer. The glass transition temperature (Tg) and the thermal effects accompanying the processes taking place during heating or cooling of the tested substance by the Differential Scanning Calorimetry (DSC) were investigated. The insoluble polymer fraction in tetrahydrofuran (THF) was used to calculate the gel content, and the soluble fraction was used to determine the average molecular weight of the sol by gel permeation chromatography (GPC). The adhesive performance was assessed by measuring adhesion, peel and shear strength. Mechanical properties were obtained by calculating the shear modulus and determining the maximum stress and strain energy. In addition, an index of glue performance was developed. A number of studies have been carried out, which have been exhaustively described and analyzed.

The article lacks a clearly defined justification for the conducted research. What are the obtained test results for? If they are to decide on a specific application, I am asking the authors to propose this application. Authors should also update the literature. Admittedly, PSA-type adhesives have been used for over half a century, but research on them is still valid. Therefore, it is worth writing what research is being carried out today. After making the above mentioned changes, I recommend this article for publication in POLYMERS.

Author Response

Dear Reviewer #3,

Firstly, thank you very much for your comments. We have read them carefully and have tried to address all the recommendations proposed. Here is a brief summary of what has been changed according to your comments.

Comment 1:The article lacks a clearly defined justification for the conducted research. What are the obtained test results for? If they are to decide on a specific application, I am asking the authors to propose this application.”

The justification of the article has been rewritten from the line 90 to 98.

Comment 2:Authors should also update the literature. Admittedly, PSA-type adhesives have been used for over half a century, but research on them is still valid. Therefore, it is worth writing what research is being carried out today.”

The literature has been updated.

Reviewer 4 Report

A lot of work is done in this paper. However, the paper is very tough to read without any chemical structures and reaction schemes.

  1. Line 12 – What properties are of interest?
  2. Line 14- What polymers are studied?
  3. Line 16- What types of polymerization?
  4. Line 95- What are A and B polymers?
  5. Introduction section must have the chemical structures of the polymers studied.
  6. Reaction schemes should be included.
  7. If GPC was used only on the soluble parts, then the molecular weight does not represent the whole polymeric system. Please clarify.
  8. Alternatively, the mobile phase in GPC should have been a solvent in which the polymer is fully soluble. Please clarify.
  9. It is not clear if a crosslinked system is formed.

Author Response

Dear Reviewer #4,

Firstly, thank you very much for your comments. We have read them carefully and have tried to address all the recommendations proposed. Here is a brief summary of what has been changed according to your comments.

Comment 1:Line 12 – What properties are of interest?”

The properties of interest are the adhesion properties. They have been indicated in the sentence “A series of Pressure-Sensitive Adhesives (PSA) was prepared using a constant monomeric composition and different preparation processes to investigate the best combination to obtain the best balance between peel resistance, tack and shear resistance” in line 12.

Comment 2:Line 14- What polymers are studied?”

The studied polymers were two water-based acrylic polymers, named A and B. The rest of the adhesive samples studied were combinations 1:1 of these two polymers.

Comment 3:Line 16- What types of polymerization?”

The following sentence has been included in line 16: “Two different strategies were applied to prepare the adhesives: Physical blending of polymers A and B, and in situ emulsion polymerization of A+B, either in one or in two steps; in this last case by polymerizing A or B first”.

Comment 4:Line 95- What are A and B polymers?”

A and B refer to the studied samples of water-based acrylic polymers. The sentence “the monomeric composition was a 1:1 combination of the two different water-based acrylic polymers, A and B” has been included in line 95.

Comments 5-6:Introduction section must have the chemical structures of the polymers studied and reaction schemes should be included”

Both chemical structures of the polymers and reaction schemes proposed have been included in the introduction section.

Comment 7: “If GPC was used only on the soluble parts, then the molecular weight does not represent the whole polymeric system. Please clarify”.

The following sentence has been included in line 270: “The average molecular weight determined by GPC only correspond to the fraction of the polymer solubilized in THF. Therefore, the values obtained should just be considered as the molecular weight of the polymer without the insoluble fraction in THF, that is, without the gel content, that cannot be analyzed by GPC”.

Comment 8:Alternatively, the mobile phase in GPC should have been a solvent in which the polymer is fully soluble. Please clarify.”

The following sentence has been included in line 141: “Tetrahydrofuran (THF) was used as a solvent to determine the gel content as it is the most suitable solvent to dissolve the acrylic polymers studied here. Accordingly, it was also used as the mobile phase in the Gel Permeation Chromatography (GPC) measurements.”

Comment 9:It is not clear if a crosslinked system is formed.”

The following sentence has been included in line 202: “The studied acrylic adhesives are thermoplastic polymeric materials with a gel fraction distributed in form of small gel domains in the bulk. According to the literature, these gel domains consist of cross-linked polymer and entangled high molecular weight polymer chains”.

Reviewer 5 Report

The present paper discusses the adhesive performance of acrylic PSA from the different preparation processes.
The paper is well written
The figures and tables are clear.
This reviewer does not have any other comments being this topic out of his expertise..

Author Response

Dear Reviewer #5,

Thank you very much for your time and comments.